# The Role of Vitamin D in the Pathogenesis of Inflammatory Bowel Disease

**Stefano Nobile [1],[*]** , **Michela A. Tenace [1]** and **Helen M. Pappa [2]**

[1]   Department of Mother and Child Health, Salesi Children's Hospital, via F. Corridoni 11, 60123 Ancona, Italy; michela.annatenace@libero.it

[2]   Division of Pediatric Gastroenterology and Hepatology, SSM Health Cardinal Glennon Children's Hospital, Saint Louis University, St. Louis, MO 63104, USA; helen.pappa@health.slu.edu

[*]   Correspondence: stefano.nobile@ospedaliriuniti.marche.it; Tel.: +39-71-596-2014

**Abstract:** Vitamin D has a complex role in the pathogenesis of inflammatory bowel disease (IBD), which is still under investigation. We conducted a literature search using PubMed through December 2018 through the use of relevant search terms. We found an abundance of evidence to support the role of vitamin D in regulating the innate and adaptive arms of the immune system. The pathogenesis of IBD implicates the immune dysregulation of these immune system components. Proof of concept of the vitamin's role in the pathogenesis of IBD is the mapping of the vitamin D receptor in a region of chromosome 12, where IBD is also mapped, and specific VDR polymorphisms' link to IBD phenotypes. Further research is needed to better delineate vitamin D's role in preventing IBD and its potential as a therapeutic target for this disease.

**Keywords:** vitamin D; inflammatory bowel disease; pathogenesis; immune system

## 1. Introduction

Vitamin D has important roles in the regulation of bone metabolism and homeostasis; nevertheless, emerging evidence highlights its role in immune regulation. Vitamin D insufficiency (serum 25OHD concentration ≤20 ng/mL) and deficiency (≤15 ng/mL) are frequently found in healthy adults and adolescents, particularly in the northern hemisphere [1]. A higher prevalence of vitamin D insufficiency and deficiency among adults and children with inflammatory bowel disease (IBD) has been reported [2–5], despite the existence of controversies [6,7]. The evaluation of vitamin D status and skeletal health in pediatric IBD has been recommended, and the promotion of physical activity may be an important measure in enhancing bone mineral density [8–11]. Diminished exposure to sunlight, decreased oral intake, nutrient malabsorption, intestinal inflammation, and gastrointestinal losses could be responsible for hypovitaminosis D in patients with IBD [7,11–13]. However, there is growing evidence that vitamin D may play a significant role in the pathogenesis of IBD.

Inflammatory bowel disease (IBD) comprises idiopathic, relapsing inflammatory diseases of the gastrointestinal tract. The pathophysiology of IBD is not completely understood, but it includes a complex interplay between gut microbiota, immune system, and environmental factors in genetically predisposed individuals.

The purpose of this paper is to review the role of vitamin D in the pathogenesis of inflammatory bowel disease.

## 2. Results

### 2.1. Vitamin D Physiology and Metabolism

Vitamin D includes a group of liposoluble hormones, such as ergocalciferol (vitamin D2) and cholecalciferol (vitamin D3). Sufficient skin exposure to ultraviolet B (UVB) radiation can satisfy the human requirement for vitamin D. Adequate vitamin D requirements may differ depending on age and morbidities: optimal vitamin D levels were associated with an intake of 400–800 IU/day in the general population [14]; specific recommendations for some diseases/conditions exist or are being evaluated, but the relative discussion is beyond the scope of this review.

In the skin, 7-dehydrocholesterol absorbs UVB radiation and it is converted to pre-vitamin D3 (pre-D3). Pre-D3 is then thermally converted to vitamin D3. Factors that are associated with cutaneous production of vitamin D3 include season, latitude, time of day, skin pigmentation, aging, and sunscreen [4]. Newly produced vitamin D3 formed in the skin, and ingested vitamin D2 or D3 (i.e., oily fish, egg yolk, supplements) binds to the plasma transport protein, vitamin D-binding protein (DBP), in the capillary bed of the dermis and the intestinal epithelium. DBP enters the circulation and then delivers vitamin D2 or D3 first to the liver, where it undergoes 25-hydroxylation, and then to the kidney, where it undergoes 1-hydroxylation [15]. 1,25-dihydroxy-vitamin D (1,25(OH)2D) is the active metabolite of the vitamin, whose action affects several (>30) tissues [16]. 1,25(OH)2D binds to the cellular vitamin D receptor (VDR), which then intracellularly migrates, producing an intracellular signaling cascade with resulting effects on various cytokine pathways. VDR expression occurs in several cells, including immune cells [17]. The most abundant metabolite in the human body, measured to assess vitamin D status, is 25OHD [15].

### 2.2. Inflammatory Bowel Disease

IBD is characterized by variable degrees of inflammation of the small bowel and/or colon, thereby presenting several clinical characteristics, including recurrent abdominal pain, gastrointestinal bleeding, diarrhea, and extraintestinal manifestations. Subtypes of this heterogeneous group of disorders are Crohn's disease (CD) and ulcerative colitis (UC). While CD can affect the entire gastrointestinal tract, from mouth to anus, the most frequently involved parts are the terminal ileum and colon. Inflammation in CD is usually transmural and often discontinuous, resulting in patchy inflammatory lesions in the gastrointestinal (GI) tract. Ulcerative colitis (UC) is mostly restricted to the rectum, colon, and cecum. UC involves continuous mucosal inflammation that is confined to the colon [17].

Studies have shown a north-south gradient for IBD, as patients who live in northern regions, with low sunlight exposure, have a higher incidence of IBD and possibly lower vitamin D levels. However, vitamin D levels may just represent a marker of sunlight exposure and they may vary according to inflammation (i.e. C reactive protein level) and albumin levels [17].

Immunological investigations showed that Crohn's disease (CD) is a predominately Th1 and Th17 mediated process, whereas ulcerative colitis (UC) is predominately mediated by Th2 and NK T-cells [18]. Several cytokines, such as IFN-$\gamma$, IL-12, and TNF-$\alpha$ are increased in CD.

Immune cells express VDRs and the enzymes that are necessary to convert vitamin D3 (25(OH)D) into 1,25(OH)2D; it has been reported that locally produced 1,25(OH)2D can exert specific autocrine and paracrine effects without producing systemic effects [19]. 1,25(OH)2D can modulate the adaptive immune response by altering the actions of activated T and B cells, and it can also modulate the innate immune response by regulating macrophages and dendritic cells maturation and function [19]. Thus, the role of vitamin D and its receptor appears to be of paramount importance for the pathogenesis of immune mediated conditions, such as IBD; we will review and present available evidence in this regard.

*2.3. Vitamin D and Immune Function*

Vitamin D3 has important immunoregulatory roles in the inflammatory and inhibitory markers of inflammatory bowel diseases.

When an antigen binds to the receptor site of antigen presenting cells (APCs: dendritic cell, macrophages, monocyte), the APCs stimulate naïve T-cells. Upon stimulation, the naïve T-cells differentiate into effector T-cells and regulatory T-cells (T-reg). The effector T-cells, under the influence of cytokines, such as IL-12, IL-4, and IL-23, further differentiate into Th1, Th2, and Th17. Th1 and Th17 cells have pro-inflammatory activities, whereas the Th2 subsets produce inhibitory cytokines. Uncontrolled production of pro-inflammatory cytokines from Th1 and Th17 subsets of T-cells is crucial for the development of various autoimmune conditions, such as inflammatory bowel diseases [18].

Group 3 innate lymphoid cells (ILC3) are tissue resident innate lymphocytes, which functionally act as Th17/22 cells in an adaptive immune system. In the healthy intestine, the most prevalent ILC3 type is IL-22-expressing natural killer cells, while IL-17- and IFN-$\gamma$-producing cells (in addition to ILC1) accumulate in the inflamed gut [20]. Konya et al. reported that IL-23 plus IL-1$\beta$ render human ILC3 responsive to vitamin D by upregulating VDR [21]. They found that vitamin D reprograms ILC3 by downregulating the IL-23R pathway while also promoting the production of IL-1$\beta$-inducible cytokines. Thus, 1,25D may serve a therapeutic agent that inhibits the IL-23R pathway in IBD [21].

Binding of Lipopolysaccharide (LPS) to Toll-like receptor 4 (TLR4) on monocytes leads to the activation of mitogen-activated protein kinase (MAPK). MAPK are critical regulators of pro-inflammatory cytokine production, including IL-6 and TNF-$\alpha$. The finding (among others) that treatment of LPS-stimulated human blood monocyte with 1,25(OH)2D was found to inhibit the release of IL-1$\alpha$, IL-6, and TNF-$\alpha$ through the up-regulation of MAPK phosphatase-1 highlights the potential therapeutic effect of vitamin D [22].

Antigen presenting cells, such as dendritic cells (DCs), are also important in inducing CD4+ T cell responses. Vitamin D3 may inhibit monocyte differentiation into DCs, and also maturation and immunostimulatory activity of DCs in vitro by inhibiting IL-12 production from DCs and upregulating IL-10 production [23,24]. Therefore, the stimulation of T-cell proliferation by DCs is inhibited. In vitro studies showed that vitamin D3 is one of the most powerful blockers of DC differentiation and IL-12 secretion, with resulting inhibition of T cell activation [25,26].

Macrophages are important APCs that clear apoptotic or senescent cells, and they are involved in repairing and remodeling tissue during wound healing. Impaired macrophage tolerance against dietary antigens and commensal microbiota is an important factor in the pathogenesis of IBD [27]. The two major subtypes are M1 and M2. M1 activity inhibits cell proliferation and it is associated with tissue damage, whereas M2 activity may enhance cell proliferation and facilitate tissue repair. Dionne et al. investigated the role of vitamin D on macrophage function and found that the M1 and M2 markers were not differentially modulated by 1,25D [28]. In their study, macrophages from CD patients were compared to those from controls, and no differences at the functional or phenotypical level were demonstrated. 1,25D administration decreased the production of pro-inflammatory cytokines by M1 cells, but it did not facilitate polarization to the anti-inflammatory M2 phenotype [28]. In contrast, Zhu et al. reported that the administration of 1,25(OH)2D reduces miR-125b expression and M2 macrophage [29]. Furthermore, 1,25(OH)2D pretreatment ameliorated colitis by restoring the lamina propria macrophage subtype balance [29].

Vitamin D may control TLR4-mediated inflammation in autoimmune diseases through the regulation of microRNAs (miRNAs). MicroRNAs are small noncoding RNAs that control gene expression. In particular, miRNA-155 has been shown to regulate the innate immune responses and TLR signaling through the regulation of several factors, including proteins that are involved in LPS signaling, such as the Fas-associated death domain protein (FADD) and IkappaB kinase epsilon (IKKepsilon) [30].

In IBD, the increased production of IFN-$\gamma$ and TNF-$\alpha$ suppresses the epithelial barrier function, resulting in intestinal mucosal permeability [31]. TNF-$\alpha$ is a central cytokine in the pathogenesis of

inflammatory bowel disease and vitamin D3 has been shown to target these inflammatory pathways in vitro, indicating a potential therapeutic role of vitamin D in IBD [32].

Another mechanism of action that is shown by vitamin D3 in Crohn's disease is the enhanced ability of T cells to upregulate programmed death-1 receptor (PD-1) and the reduction of CD69 expression—a marker of T cell activation [33]. Activated T cells, B cells, and Natural Killer cells express PD-1 receptor, and it promotes immune tolerance through different mechanisms, such as the inhibition of T cell function, survival, and activation.

Alterations of the fecal microbiome may be another mechanism through which vitamin D influences the risk of developing IBD, as shown by preliminary studies in rat models [34]. In a small interventional study, Schäffler et al. found that, in contrast to healthy controls, the administration of vitamin D to patients with CD was associated with changes in bacterial intestinal flora, which are particularly evident after one week: the typical bacteria shifted from Betaproteobacteria to Bacteroidetes [35]. However, a further increase of the vitamin D level over three weeks was associated with a reversal of this effect and with a decreased bacterial diversity in the CD microbiome [35]. The authors could not explain the mechanisms for the observed shift in bacterial communities and change in microbiota diversity; they hypothesized that there could be an optimal vitamin D "window" of administration and dose that may be beneficial for CD patients.

## 2.4. VDR Gene and IBD

The VDR gene is located in a chromosome 12 region that has been linked to IBD by genome screening techniques. Genetic variation might alter the binding affinity of the vitamin D VDR and it was associated with immune and inflammatory disease, including IBD [36].

Vitamin D and VDR are thought to protect the intestine from damage by several mechanisms, which can be grouped into two main chapters: maintenance of epithelial barrier function and decreased mucosal inflammation [37]. Du et al. showed that 1,25(OH)2D3-VDR signaling preserves the mucosal barrier integrity by halting tight junction dysregulation due to myosin light chain kinase (MLCK) in a cell model of active ulcerative colitis [38]. Increases in tight junction permeability occur in early disease stage and promote disease initiation, whereas disease progression may depend on tight junction-independent barrier loss (i.e., due to epithelial cell apoptosis). Other studies showed that, in intestinal biopsies from IBD patients, the mucosal VDR levels are reduced, which is probably due to TNF-$\alpha$-induced increase of miR-346 (a factor influencing VDR translation in epithelial cells) [39–41]. In another experimental study, gut epithelial VDR deletion resulted in impaired epithelial cell apoptosis and increased mucosal barrier permeability; eventually, invading luminal bacteria activated mucosal TH1 and TH17 responses [42]. Furthermore, Stio et al. reported that claudin-1 and claudin-2 proteins (tight junction proteins that are involved in the regulation of paracellular permeability) were up-regulated in active UC [43]. The administration of 1,25(OH)2D3 decreased claudin-1 and claudin-2 protein levels either in inflamed and non-inflamed biopsies, suggesting a potential role of vitamin D for the treatment of active UC [43]. Zhang et al. showed that Claudin-2 gene is a direct target of transcription factor VDR (Claudin-2 was up-regulated by over-expressed VDR), and that, in biopsies from active UC patients, VDR expression was low and Claudin-2 was increased [44,45]. Increased Claudin-2 was associated with more severe intestinal leakage, enhanced permeability, and inflammation in vitamin D receptor knockout mice.

In colonic mucosal samples from IBD patients as compared to controls, VDR expression was down-regulated, whereas vitamin D1-$\alpha$ hydroxylase (CYP27B1) expression was increased [46]. Experimental studies showed that inactivation of the VDR in macrophages and granulocytes resulted in increased pro-inflammatory cytokine production, although marginally affecting colitis-associated symptoms [47].

In a small case-control study, Abreu-Delgado et al. studied the relationship between serum vitamin D levels, mucosal vitamin D receptor (VDR) expression, and histologic disease activity [48].

They reported that vitamin D levels were positively associated with mucosal VDR expression in non-inflamed mucosa and that colonic VDR expression was decreased in inflamed mucosa.

With regard to VDR polymorphisms, several studies have highlighted associations between specific polymorphisms and IBD. Xue et al. (2013) performed meta-analyses and examined VDR polymorphisms (Taql, Bsml, Fokl, and Apal) in different populations [49]. The authors found that: in Asians, the Fokl ff genotype was associated with an increased risk of UC (OR = 1.65; 95% CI, 1.11–2.45) and that the "a" allele of the Apal polymorphism was protective against CD (OR = 0.81; 95% CI, 0.67–0.97); in Europeans, the Taql tt genotype was associated with increased risk of CD (OR = 1.23; 95% CI, 1.02–1.49) and (in males) with increased risk of UC (OR = 1.56; 95% CI, 1.02–2.39) and CD (OR = 1.84; 95% CI, 1.19–2.83) [49].

Simmons et al. (2000) compared Taql polymorphism between 158 UC patients, 245 CD patients, and 164 cadaveric renal allograft donor controls and found a higher frequency of the tt genotype in CD patients (frequency 0.22) when compared to UC patients (0.12) or controls (0.12) (odds ratio 1.99; 95% confidence interval [CI] 1.14–3.47; $p$ = 0.017) [50].

In another study that was conducted in an Ashkenazi Jewish population in which the Bsml VDR polymorphism was assessed, the frequency of the BB genotype was higher in UC patients than the controls (0.21 vs. 0.11 $p$ = 0.042, odds ratio 2.27, (95% confidence interval [CI] 1.06–4.9) [51]. Other authors found a probable association with the ff genotype of the Fokl polymorphism ($p$ < 0.001 in Iranian CD patients) [52].

In a Korean case-control study and meta-analysis that aimed at evaluating the association between VDR FokI polymorphisms and colorectal diseases, Cho et al. reported increased risk of IBD among individuals carrying the FokI f allele when compared to those carrying the F allele [53].

Other studies reported a correlation between VDR polymorphisms and clinical characteristics of IBD. For example, Gisbert-Ferrándiz et al. found that CD patients who were homozygous for a single nucleotide polymorphism (SNP)—rs731236—presented with lower VDR protein levels in circulating mononuclear cells, increased IL1β mRNA levels, and enhanced expression of lymphocytic adhesion molecules [54]. These patients had a higher risk of developing the peculiar penetrating CD phenotype and a higher risk of IBD-related surgery. Zheng et al. examined the association between VDR polymorphisms and the serum level of 25-hydroxyvitamin D in a Chinese Han population of UC and controls; they found a significant correlation between FokI polymorphism and vitamin D deficiency (<20 ng/mL) in UC patients, and between such polymorphism and UC severity of disease [55]. In contrast, BsmI polymorphism and the frequency of the AAC haplotype formed by BsmI, ApaI and TaqI were significantly lower in UC compared with controls.

VDR is also a transcription factor. The target genes of VDR include anti-microbial peptide (AMPs) cathelicidin antimicrobial peptide, β-defensin, and the 1,25(OH)2D3-regulated VDR-specific, Cyp24 hydroxylase gene. VDR deletions exacerbate colitis through activation of the NF-κB pathway. Wu et al. showed that VDR is an important regulator of intestinal homeostasis and it is involved in several functions, such as autophagy, intestinal microbiome composition and variations, and innate immunity (particularly Paneth cells)—all of the factors that have been implicated in the pathogenesis of IBD [56]. There are important interactions between vitamin D, VDR, and intestinal microbiome that still have to be fully elucidated.

Importantly, a reduced level of VDR expression in colonic mucosa in UC has been associated with an increased risk of carcinogenesis [57]. Thus, it is likely that VDR may represent a marker of mucosa dysplasia and cancer in UC patients, and that it could potentially be useful in patients surveillance and tumor prevention.

## 2.5. Vitamin D binding Protein

Free 25OHD is transported in the blood that is mainly linked to the vitamin D Binding Protein (VDBP) (85% to 90%). The VDBP gene is subject to polymorphism, leading to variants with differing affinity for both 25(OH)D and 1,25(OH)2D [58]. The VDBP variants have been consistently correlated

with circulating 25(OH)D and VDBP concentrations. Furthermore, VDBP has unique roles in actin scavenging and in neutrophil chemotaxis, and it is less subject to seasonal variation when compared with total 25(OH)D. Thus, investigating VDBP is important when considering the effect of vitamin D metabolism in health and disease.

Some authors found that VDBP levels were inversely correlated with inflammation in pediatric IBD, and it is speculated that the underlying reasons were reduced liver production and/or intestinal losses [5].

Ghaly et al. found that higher VDBP concentrations were significantly associated with IBD flare, independent of season, sex, age, ethnicity, treatment type, smoking status, and mode of remission induction [59].

In a Swiss population, Eloranta et al. compared the frequency of two single nucleotide polymorphisms (SNp) of Vitamin D Binding protein (DBP) between IBD patients and controls, and they found that the DBP 420 variant Lys was less frequent in IBD cases. Moreover, the DBP 416 polymorphism was not associated with IBD, and the haplotype that consisted of 416 Asp and 420 Lys was more frequent in the controls than in UC patients [60].

## 3. Discussion

Vitamin D is a pleotropic hormone with a pivotal role in the regulation of the immune system. Its role in the pathogenesis of IBD is probably complex and involves several aspects of the immune system:

- the innate immune system, where vitamin D promotes balance of macrophage subtypes, inhibits excessive stimulation of T-cells by dendritic cells, and promotes epithelial barrier integrity [23–31,38–45]; and,
- the adaptive immune system, where vitamin D downregulates proinflammatory pathways, such as the IL-23R pathway and the TNF-$\alpha$ and IFN-$\gamma$ pathways [21–23,26–28,33].

It is known that the disruption in the balance between regulatory and proinflammatory pathways of the immune system, as well as a compromise in the epithelial barrier integrity, is the basis for the pathogenesis of IBD, thus underlying the importance and the potential role of vitamin D in this disorder. Vitamin D exerts other important effects on intestinal microbiome [34,35], and further research is needed to better elucidate this issue.

As proof of concept for the role of vitamin D in the pathogenesis of IBD, the vitamin D receptor gene maps to a region of chromosome 12, which is also linked to IBD [36]. Moreover, polymorphisms of the vitamin D receptor have been linked to IBD phenotypes, pointing toward a relationship between altered affinity of vitamin D to its receptor and IBD [49–55].

Further research is warranted to examine whether deficiency of the circulating vitamin D metabolites, a disruption of any of the steps in vitamin D homeostasis, or a defect in its interaction with its receptor could play a role in immune dysregulation, leading to the pathogenesis of IBD. Furthermore, additional research may uncover actions of Vitamin D, thereby leading to novel therapeutic targets and therapies (less toxic than current).

## 4. Materials and Methods

We conducted a literature search using PubMed through December 2018. We used the following search terms: "inflammatory bowel disease", "ulcerative colitis" and "Crohn's disease", "Vitamin D", "Vitamin D receptor", "Vitamin D binding protein", "Vitamin D receptor polymorphism", "inflammatory cytokines", and "immune response". The search terms included both MeSH terms and text words. To identify any additional studies, we also screened the references of the retrieved publications. This search was limited to human studies and publications written in English. We did not consider abstracts or unpublished reports. The search was conducted using these terms in the keywords, abstracts, and titles.

The aims of this review were to assess the role of vitamin D on immune cells, intestinal epithelium and cytokine profile in patients with IBD, and to describe VDR associations, polymorphisms, and mechanism of action in this population.

**Author Contributions:** Conceptualization, S.N.; Methodology, S.N., M.A.T., H.M.P.; Writing—Original Draft Preparation, S.N., M.A.T.; Writing—Review & Editing, S.N., H.M.P.

**Funding:** This research received no external funding.

**Conflicts of Interest:** The authors declare no conflict of interest.

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
