# Peer review of "The Role of Vitamin D in the Pathogenesis of Inflammatory Bowel Disease"

_gastrointestdisord, doi:10.3390/gidisord1010018_

Round 1

Reviewer 1 Report

Minor revision are required as follows

Some other typing errors must be revised through all the manuscript, for example: lane 28 page 1 ‘’controveries’’ must be corrected in ‘’controversies’’; lane 25 page 1 ‘’?’’ must be replaced with the sign ‘’<’’ and so on in the manuscript the correct replacement of ‘’?’’ with proper symbols is required.

Page 2, from lane 58 to lane 66: a reference is required.

Page 3, lane 134: another mechanism […] ‘’is’’ and not ‘’are’’

Discussion could be improved, and references are required.

Author Response

We thank the reviewer for his/her valuable comments. We have corrected typos and grammatical errors and tried to address all of the comments/suggestions.

Reviewer 2 Report

This is a great review of the role of vitamin D,its receptors, and polymorphisms in the potential pathogenesis of IBD.  

There are some changes to the manuscript that is recommended before considering for publication:

I have attached specific comments in the pdf and some potential changes to the wording/ consolidation of some sentences for ease of reading.  There were several spelling changes as well as "?" next to symbols that should be corrected for the manuscript.

Additionally,

On page 2, first paragraph, please include how much UVB radiation is "sufficient" as duration of exposure to satisfy the human daily requirement for vitamin D.  Since it is a review, please also include the actual daily requirement level.  Is this different for the IBD patient or a patient with immune related diseases compared to the general population?

Page 3:  line 126:  microRNSs should be spelled microRNAs.  

Page 4, first paragraph (lines 143-145).  If overall the evidence suggests that lower vitamin D levels contribute to IBD, can the authors explain which the second study [ref 34] shows that further increase of vitamin D levels are linked to a decreased bacterial diversity in CD?  In general we understand that decreased bacterial diversity is detrimental in IBD.  

Author Response

We thank the reviewer for his/her valuable comments. We have corrected typos and grammatical errors and tried to address all of the comments/suggestions. We briefly discussed vitamin D requirement and provided a reference. With regard to the last point, Schaffler et al could not explain the mechanisms for the observed shift in bacterial communities and change in microbiota diversity; they hypothesized that there could be optimal vitamin D “window” of administration and dose that may be beneficial for CD patients.

Reviewer 3 Report

In this paper, the authors review the relationship between vitamin D and IBD. It is very attractive and important theme in nutritional and medical fields. It was a rare delight to read a manuscript in which I found nothing to fix about the content. This is a well-written, but there are some garbled in this manuscript (for example; line 25 and 73). After correcting them, the reviewer accepts the acceptance of this paper.

Author Response

We are sincerely grateful to the reviewer for her/his comment and fixed the typos throughout the manuscript.

Round 2

Reviewer 2 Report

Thank you for making the appropriate changes.